# Comprehensive Analysis of *YTH Domain-Containing* Genes, Encoding m^6^A Reader and Their Response to Temperature Stresses and *Yersinia ruckeri* Infection in Rainbow Trout (*Oncorhynchus mykiss*)

**DOI:** 10.3390/ijms24119348

**Published:** 2023-05-27

**Authors:** Han Yu, Qinfeng Gao, Wen Wang, Dazhi Liu, Jinghong He, Yuan Tian

**Affiliations:** 1Key Laboratory of Mariculture, Ministry of Education, Ocean University of China, Qingdao 266003, China; yh9500@stu.ouc.edu.cn (H.Y.); qfgao@ouc.edu.cn (Q.G.); ldz@stu.ouc.edu.cn (D.L.); hejinghong@stu.ouc.edu.cn (J.H.); 2Function Laboratory for Marine Fisheries Science and Food Production Processes, Qingdao National Laboratory for Marine Science and Technology, Qingdao 266100, China

**Keywords:** rainbow trout, *YTH domain-containing* genes, m^6^A reader, temperature stress, immune challenge

## Abstract

*YTH domain-containing* genes are important readers of *N^6^*-methyladenosine (m^6^A) modifications with ability to directly affect the fates of distinct RNAs in organisms. Despite their importance, little is known about *YTH domain-containing* genes in teleosts until now. In the present study, a total of 10 *YTH domain-containing* genes have been systematically identified and functionally characterized in rainbow trout (*Oncorhynchus mykiss*). According to the phylogenetic tree, gene structure and syntenic analysis, these *YTH domain-containing* genes could be classified into three evolutionary subclades, including YTHDF, YTHDC1 and YTHDC2. Of them, the copy number of *OmDF1*, *OmDF2*, *OmDF3*, and *OmDC1* were duplicated or even triplicated in rainbow trout due to the salmonid-specific whole-genome duplication event. The three-dimensional protein structure analysis revealed that there were similar structures and the same amino acid residues that were associated with cage formation between humans and rainbow trout, implying their similar manners in binding to m^6^A modification. Additionally, the results of qPCR experiment indicated that the expression patterns of a few *YTH domain-containing* genes, especially *OmDF1b*, *OmDF3a* and *OmDF3b*, were significantly different in liver tissue of rainbow trout under four different temperatures (7 °C, 11 °C, 15 °C, and 19 °C). The expression levels of *OmDF1a*, *OmDF1b* and *OmDC1a* were obviously repressed in spleen tissue of rainbow trout at 24 h after *Yersinia ruckeri* infection, while increased expression was detected in *OmDF3b*. This study provides a systemic overview of *YTH domain-containing* genes in rainbow trout and reveals their biological roles in responses to temperature stress and bacterial infection.

## 1. Introduction

*N^6^*-Methyladenosine (m^6^A), occurring in the sixth N atom of adenosine residues, is the most prevalent post-transcriptional modification in all eukaryotic coding and non-coding RNAs [1,2,3]. Recently, accumulating evidence has revealed that m^6^A modification could heavily affect the various aspects of RNA processing and metabolism, including alternative splicing, alternative polyadenylation, nuclear export, subcellular localization, mRNA stability, and translation control [4,5,6,7]. Therefore, it is proposed to be crucial for a variety of biological processes in eukaryotes, such as cell cycle regulation, embryonic development [8], sex determination [9], neuronal functions [9], and cancer progression [10]. Additionally, RNA m^6^A modifications are also necessary to recover from biotic and abiotic stresses [11,12].

Similar to DNA methylation, the functions of RNA m^6^A modifications are dependent on their recognition by specific RNA binding proteins, also known as m^6^A readers [13]. At present, several kinds of m^6^A readers have been identified in higher vertebrates, among which the best characterized members are *YT521-B homology* (*YTH*) *domain-containing* genes, namely YTHDF1-3, YTHDC1 and YTHDC2 [14]. These genes can selectively recognize and bind to m^6^A modifications of RNAs by using an aromatic cage in YTH domain, which forms a hydrophobic pocket around the modified nucleotide [14,15,16].

It is well documented that the m^6^A binding capability of *YTH domain-containing* genes is relatively conserved in eukaryotic evolution. The binding site of *YTH domain-containing* genes in human (*Homo sapiens*) is the DRACH motif. (D = A/G/U, R = A/G and H = A/C/U) [17,18]. The aromatic cage-like pocket of YTHDF protein in mouse (*Mus musculus*) is composed of tryptophan, tryptophan and tryptophan (WWW) [19]. Meanwhile, the pocket of YTHDC protein in mouse is composed of tryptophan, tryptophan, and leucine (WWL) [19]. In addition, there are also diversities and multiplicities among the three evolutionary subclades (YTHDF, YTHDC1 and YTHDC2), offering a multitude of possible combinations that can rapidly and finely regulate gene expression patterns and enhance the cellular plasticity in response to various cues [20,21]. In general, YTHDF1-3 has been mainly associated with RNA translation and degradation. Emerging studies have revealed that the main role of YTHDF1 is to recruit the eukaryotic translation initiation factor 3 (eIF3) and promote the translation of m^6^A mRNA [22]. The main function of YTHDF2 could service as an important regulator of RNA stability by interacting with the CCR4/CAF/NOT complex [23]. YTHDF3 is able to contribute to either enhancing the translation or triggering the degradation of m^6^A RNA [24]. In contrast, YTHDC1 has been identified as being required for some alternative splicing as well as alternative polyadenylation events of pre-mRNAs by recruiting the distinct splicing factors or RNA processing regulators [25]. YTHDC2 is not only involved in translation enhancement, but also implicated in degradation induction [26,27]. 

Because of their importance, *YTH domain-containing* genes have been systematically defined and extensively studied in higher vertebrates in the last decade. For example, by recognizing the m^6^A modification of key genes in p38, AKT, ERK1/2, and NF-κB pathways, *YTHDF2* could take part in regulating cell injury and apoptosis in the lung bronchial epithelial tissue of humans [28]. YTHDF1 is able to promote functional axon regeneration in the peripheral nervous system by facilitating the global translation of injury-induced proteins in dorsal root ganglion of adult mouse [29]. In cattle (*Bos taurus*), YTHDF1 could recognize the m^6^A modification in the 3′-UTR region of creatine kinase, promote its mRNA translation or stabilization to stimulate the differentiation of bovine skeletal myoblast and regulate the process of myogenesis [30,31]. Additionally, *YTH domain-containing* genes also play important roles in response to biotic and abiotic stresses. In humans, *YTHDF2* is proven to participate in the regulation of the gene expression of *HSP90*, *HSP60* and *HSPB1* in response to heat shock stress [32]. It has been reported that the heat shock-inducible lncRNA acts as a transcriptional brake to restrain the gene expression of heat shock transcription factor 1, and it takes part in regulating the heat shock response in mice. However, the heat shock-inducible lncRNA is mediated by the nuclear m^6^A reader YTHDC1 [33]. In the liver of sheep (*Ovis aries*), the gene expression levels of *YTHDC1* and *YTHDC2* and the protein abundance of YTHDF3 and YTHDC1 are significantly changed by heat stress, which may be associated with the altered expression of several heat shock proteins (HSPs), including HSP70, HSP90 and HSP110 [34]. A similar observation is also reported in bovine animals [35]. Compared with higher vertebrates, functional studies of *YTH domain-containing* genes have remained limited in teleosts, where there have only been reports on zebrafish (*Danio rerio*) [8] and olive flounder (*Paralichthys olivaceus*) [36]. Through m^6^A-dependent binding, YTHDF2 could precisely control the massive decay of maternal mRNAs and promote the zygotic genome activation during the early life of embryos in zebrafish [8]. Both YTHDF1 and YTHDF2 exhibit high expression levels in the testis of olive flounder, which may increase the translation of methylated mRNA related to testis development [36]. Although the related studies remained limited, they could support the hypothesis that *YTH domain-containing* genes are tightly related to various physiological processes in teleosts. Hence, it is of great significance to further investigate and study *YTH domain-containing* genes in teleosts.

Rainbow trout (*Oncorhynchus mykiss*) is considered one of the most economically important and promising aquaculture species around the world [37]. In addition, rainbow trout also acts as an excellent model in many research fields, such as physiology, immunology, ecology, and toxicology [38,39,40,41,42]. However, *YTH domain-containing* genes have yet to be systematically studied up to this point. In the present study, a set of *YTH domain-containing* genes were identified in rainbow trout. The comprehensive analysis of the phylogenetic tree, synteny relationships and 3D protein structure were conducted to functionally characterize these *YTH domain-containing* genes. Additionally, the expression levels of *YTH domain-containing* genes were also determined in the liver and spleen tissues of rainbow trout under different temperatures or after *Yersinia ruckeri* infection. This study provides a comprehensive view of *YTH domain-containing* genes in rainbow trout and reveals their biological functions in response to environmental stress and immune challenges.

## 2. Results

### 2.1. Identification of YTH Domain-Containing Genes in Rainbow Trout

In the present study, a total of 10 *YTH domain-containing* genes were systematically identified and functionally characterized in rainbow trout. According to their corresponding homolog genes in model animals, these *YTH domain-containing* genes in rainbow trout were designated as *OmDF1a*, *OmDF1b*, *OmDF1c*, *OmDF2a*, *OmDF2b*, *OmDF3a*, *OmDF3b*, *OmDC1a*, *OmDC1b*, and *OmDC2*, respectively (Table 1). The detailed information of these genes were given in Table 1. In brief, the gene lengths ranged from 7206 bp (*OmDF1b*) to 41,839 bp (*OmDC1b*). Interestingly, it was found that there was disagreement between the lengths of the genes and their corresponding open reading frames (ORFs). *OmDC2* had the largest ORF with 4326 bp, followed by *OmDC1b* (2124 bp) and *OmDC1a* (2049 bp); the smallest ORF was founded in *OmDF1a* (1812 bp), the lengths of protein encoded by which varied from 603 aa (*OmDF1a*) to 1441 aa (*OmDC2*).

In addition, the prophesied molecular weights of 10 *YTH domain-containing* genes were different and ranged between 65,089.20 (*OmDF1a*) and 160,900.86 (*OmDC2*); the isoelectric points ranged from 7.82 (*OmDF1a*) to 9.45 (*OmDC2*). The subcellular localization prediction revealed that 7 *OmDF* genes were mainly distributed in the cytoplasm region, whereas *OmDC* genes located in the nuclear region. The cDNA sequences of these *YTH domain-containing* genes were submitted to GenBank database with the accession numbers listed in Table 1.

### 2.2. Phylogenetic Analysis of YTH Domain-Containing Genes

To elucidate the evolutionary relationships of *YTH domain-containing* genes, an un-rooted phylogenetic tree was constructed based on 56 full-length amino acid sequences of *YTH domain-containing* genes from 10 representative vertebrates, including human, mouse, chicken (*Gallus gallus*), zebrafish, Japanese medaka (*Oryzias latipes*), Nile tilapia (*Oreochromis niloticus*), channel catfish (*Ictalurus punctatus*), large yellow croaker (*Larimichthys crocea*), giant grouper (*Epinephelus lanceolatus*) and rainbow trout. In addition, a phylogenetic tree of several Salmoniformes species was constructed based on 81 full-length amino acid sequences of *YTH domain-containing* genes from Atlantic salmon (*Salmo salar*), brown trout (*Salmo trutta*), chinook salmon (*Oncorhynchus tshawytscha*), chum salmon (*Oncorhynchus keta*), coho salmon (*Oncorhynchus kisutch*), sockeye salmon (*Oncorhynchus nerka)*, lake trout (*Salvelinus namaycush*) and rainbow trout. As shown in Figure 1, these phylogenetic trees were divided into five major clades supported by the robust bootstrap values. These clades represented the distinct *YTH domain-containing* genes. Due to the events of salmonid-specific whole-genome duplication, almost all of the *YTH domain-containing* genes in Salmoniformes species were duplicated or even triplicated, including *YTHDF2*, *YTHDF3* and *YTHDC1*. In sharp contrast, the high vertebrates or other teleosts only have a single copy of these *YTH domain-containing* genes.

All the *YTH domain-containing* genes in rainbow trout were clearly grouped with their corresponding homologs. The results also provided strong evidence for the accuracy and reliability of gene annotation in the present study. In addition, these five clades could be further clustered into three groups, which is consistent with their subfamilies classification of YTHDF, YTHDC1 and YTHDC2, suggesting an evolutionary divergence among subfamilies of *YTH domain-containing* genes in vertebrates.

### 2.3. Gene Structure and Conserved Motif Analysis

The gene structure analysis showed that all members of *YTH domain-containing* genes in rainbow trout consisted of at least five exons and four introns (Figure 2A). The distribution of exon and intron phases were largely coincident with the clusters or clades of the phylogenetic tree of *YTH domain-containing* genes (Figure 2A). Members from same subfamilies, YTHDF or YTHDC, shared highly similar numbers of exons and introns. However, there were great divergences in the sizes and arrangements of the exons and introns between these *YTH* members from the YTHDF and YTHDC subfamilies. As shown in Figure 2B, each YTH protein contained the complete YTH domain that was composed of approximately 135 amino acid residues. In general, the YTH domain was located near the C-terminal except for *OmDC1a* and *OmDC1b*, the domain of which appeared in the middle regions.

A total of 10 conserved motifs were identified and characterized among these YTH proteins in rainbow trout, and the lengths ranged from 15 to 50. The detailed information of these conserved motifs is provided in Figure 2C. Among them, motif 1, motif 2 and motif 3 constituted the YTH domain, which was present in all of the YTH proteins (Figure 2C, Table 2). Meanwhile, the remaining motifs (motif 4–7) only existed in YTH members of YTHDFs and shared a similar distribution, indicating their conversation in evolution.

### 2.4. Chromosome Locations and Synteny Relationships of YTH Domain-Containing Genes

In rainbow trout, these *YTH domain-containing* genes were distributed on 9 chromosomes, including Chr05, Chr09, Chr11, Chr12, Chr15, Chr16, Chr19, Chr25, and ChrY (Figure 3A). It was found that almost all of the *YTH domain-containing* genes were dispersed among different chromosomes; only *OmDF1a* and *OmDF1c* were located on the same chromosome (Chr16). *OmDF1a*, *OmDF1b*, *OmDF1c*, *OmDF2a*, *OmDF2b*, *OmDF3a*, *OmDF3b*, *OmDC1a*, and *OmDC1b* were distributed in the paralogous regions originating from the salmonid-specific whole-genome duplication event. A genomic synteny analysis was performed to investigate the homology of *YTH domain-containing* genes among zebrafish, Japanese medaka, Atlantic salmon, and rainbow trout (Figure 3B). There were numerous conserved syntenic blocks between Atlantic salmon and rainbow trout, which harbored twenty-two homologous pairs of *YTH domain-containing* genes. There were diverse syntenic gene pairs among the *YTH domain-containing* genes in zebrafish, Japanese medaka, Atlantic salmon, and rainbow trout, such as *DrDF1*-*OlDF1a*-*OlDF1b-SsDF1a*-*SsDF1b*-*SsDF1c*-*OmDF1a-OmDF1b-OmDF1c*, *DrDF2*-*OlDF2-SsDF2a*-*SsDF2b-OmDF2a-OmDF2b*, *DrDF3*-*OlDF3-SsDF3a-SsDF3b-OmDF3a-OmDF3b*, *DrDC1*-*OlDC1-SsDC1a-SsDC1b-OmDC1a-OmDC1b*, and *DrDC2*-*OlDC2*-*SsDC2-OmDC2*. The results revealed the formation of duplicated *YTH domain-containing* genes from the salmonid-specific whole-genome duplication.

### 2.5. Multiple Sequence Alignments and 3D Protein Structure Analysis

Multiple alignments were performed to investigate the sequence characteristics of the functional YTH domains, the results of which clearly showed the highly conserved sequences of YTH domains between human and rainbow trout, especially several key amino acid residues that were involved in the formation of cage and binding to m^6^A modification (Figure 4). The 3D protein structures of *YTH domain-containing* genes in human genes were constructed based on the known models in the SWISS-MODEL database. Meanwhile, we also predicated these 3D protein structures in rainbow trout through homology modeling using AlphaFold. These proteins were mainly composed of six outside α-helices and eight inside β-strands, which revealed the presence of relative conservation of structures in the functional domains of *YTH domain-containing* genes between human and rainbow trout.

In addition, YTH-GG (m^6^A) CU complexes were further visualized to show the combination and interactions between the YTH readers and RNA m^6^A modification (Figure 5). The visualizations clearly showed that m^6^A modification was buried in a deep cleft that was formed by three hydrophobic tryptophan residues (cage residues), such as W439, W493 and W498 in *OmDF1a*; W446, W500 and W505 in *OmDF2a*; and W473, W527 and W532 in *OmDF3a* (Figure 5). The m^6^A modification points to rings of the three tryptophan residues and forms base-specific hydrogen bonds with the other three hydrophobic residues (H bond residues), such as Y425, D429 and C440 in *OmDF1a*; Y432, D436 and C447 in *OmDF2a*; Y459, D463 and C474 in *OmDF3a*; N311, N315 and S326 in *OmDC1a*; and S1306, N1310 and S1321 in *OmDC2* (Figure 5). Through the comparison, the YTH domains in human and rainbow trout genes exhibited a similar manner of binding to m^6^A modification.

### 2.6. Expression Patterns of YTH Domain-Containing Genes under Different Temperatures

To explore the potential functions of *YTH domain-containing* genes in response to different temperature conditions, qPCR experiments were performed to determine their expression patterns in the liver tissue of rainbow trout under four different temperature conditions, including 7 °C, 11 °C, 15 °C, and 19 °C (Figure 6). As reported, 15 °C was generally accepted as the optimum temperature for the growth of rainbow trout. Hence, it was set as control to calculate the expression levels of *YTH domain-containing* genes in qPCR experiments. Compared with the control (15 °C), low temperature significantly induced the expression of *OmDF1a*, *OmDF1b*, *OmDF1c*, *OmDF2a*, *OmDF2b*, *OmDF3a*, *OmDF3b*, and *OmDC2* (*p*-value < 0.05). Meanwhile, an up-regulated expression of *OmDF1b*, *OmDF3a* and *OmDF3b* also existed in the high temperature condition (19 °C) compared with the control at 15 °C (*p*-value < 0.05). However, there were no significant differences in the expression levels of *OmDC1a* among different temperatures (*p*-value > 0.05).

### 2.7. Expression Patterns of YTH Domain-Containing Genes after Y. ruckeri Infection

Additionally, RNA-seq datasets were performed to construct the expression patterns of *YTH domain-containing* genes in spleen tissue of rainbow trout and uncover their potential roles in response to bacterial infection (Figure 7). The results revealed that *OmDC1a* had a relatively higher abundance in spleen of rainbow trout. The expression levels of *OmDF1a*, *OmDF1b* and *OmDC1a* were obviously repressed at 24 h after *Y. ruckeri* infection. In contrast, increased expression was detected in *OmDF3b* at 24 h after *Y. ruckeri* infection, suggesting its importance in response to bacterial infection.

## 3. Discussion

RNA m^6^A modifications are crucial for the regulation of gene expression at post-transcriptional level and play important roles in response to abiotic and biotic stresses in organisms [3,44]. The recognitions of m^6^A modifications by reader proteins are pivotal for performing the multiple functions. However, the information of *YTH domain-containing* genes, which encode the major readers of m^6^A modifications, remains largely unexplored in teleosts. In the present study, *YTH domain-containing* genes were systematically identified and functionally characterized in rainbow trout. In addition, the qPCR experiment and RNA-seq datasets were performed to determine the expression levels of these *YTH domain-containing* genes in liver or spleen tissues of rainbow trout under different environmental temperatures or after *Yersinia ruckeri* infection. This study provided a comprehensive view of *YTH domain-containing* genes in rainbow trout, and laid a foundation for further characterizations of their biological functions in response to environmental stress and immune challenge.

Five *YTH domain-containing* genes have been previously identified in humans [37], and five *YTH domain-containing* genes were obtained in zebrafish [45] and olive flounder [36]. In sharp contrast, the members of *YTH domain-containing* genes in rainbow trout are increased to 10 because of the duplicated or even triplicated copies of *OmDF1*, *OmDF2*, *OmDF3*, and *OmDC1*. Similar duplications of *YTH domain-containing* genes were also observed in Atlantic salmon [46]. The divergences in copy members of *YTH domain-containing* genes among higher vertebrates, teleosts and salmonids could result from the salmon-specific 4R whole-genome duplication. It is further supported by the locations of duplicated or triplicated copies in the homologous regions of chromosomes in both rainbow trout and Atlantic salmon. More *YTH domain-containing* genes in teleosts may reflect their more complex regulatory mechanism or the functional redundancy of m^6^A readers [47,48].

Although there were differences in the copy members of *YTH domain-containing* genes among various species, the homologs were grouped together in the phylogenetic tree and displayed complete YTH domains with similar sizes of ~135 amino acid residues. This observation suggests the conservation of *YTH domain-containing* genes in evolution. Meanwhile, YTH domains in human and rainbow trout genes were both composed of six outside α-helices and eight inside β-strands and showed the same amino acid residues associated with cage formation, implying their similar manners in binding to m^6^A modification. These results provided additional evidence for the evolutionary conservation of *YTH domain-containing* genes among distinct species. Moreover, *YTH domain-containing* genes fall into three subfamilies or evolutionary subclades, namely YTHDF, YTHDC1 and YTHDC2 [3,21]. Aside from the conserved YTH domain, YTHDC1 and YTHDC2 seem to be unrelated to the other YTH *domain-containing* genes in rainbow trout, with different gene sizes, subcellular localizations, exon and intron arrangements, motif organizations, and cage residues. These results are largely consistent with previous observations in higher vertebrates, such as in humans [49] and mouse [50]. The results demonstrate that there seem to be functional divergences among the subfamilies or evolutionary subclades of *YTH domain-containing* genes.

Temperature is an important environmental factor that directly affects the survival, growth and development of aquatic organisms such as fish, and it also affects some other physiological processes [51,52]. Emerging evidence suggests that m^6^A acts as a dynamic modification with important roles in response to environmental stresses, the multiple functions of which are depended on reader proteins, encoded by *YTH domain-containing* genes [37,53]. In the present study, the expression patterns of *OmDF1b*, *OmDF1c*, *OmDF2a*, *OmDF2b*, *OmDF3a*, *OmDF3b*, *OmDC1b*, and *OmDC2* in rainbow trout were dramatically changed by different environmental temperatures. In embryonic fibroblasts of mouse, expressions of YTHDF1-3 were markedly increased after heat stress, which is in line with the observations at 19 °C in the present study [54]. YTHDF2 could participate in the regulation of heat shock genes [54]. Following a temperature increase, YTHDF2 could bind to m^6^A modifications in the 5′-UTRs of heat shock mRNAs to protect them from demethylases and accompany with eIF3 to promote their cap-independent translation in response to heat stress [21,55]. It has been proven that YTHDC1 is required for the transcription termination of heat shock proteins and subsequently attenuates the heat-shock responses [56,57]. In the present study, the repressed expression of *OmDC1b* indicated that the heat-shock responses are still needed to maintain the homeostasis of liver in rainbow trout at 19 °C. In addition, the expression levels of several *YTH domain-containing* genes were also influenced by low temperature, suggesting a significant contribution to the cellular response to cold stress. However, the biological functions of these m^6^A readers have yet to be investigated and reported in animals.

RNA-seq datasets revealed that the expression patterns of several *YTH domain-containing* genes were responsive to *Y. ruckeri* infection in spleen of rainbow trout, especially *OmDF1a*, *OmDF1b*, *OmDF3b*, and *OmDC1a*. It is believed that YTHDF1 and YTHDF3 are able to enhance the translation of target genes and facilitate inflammation by recognizing the m^6^A modification against bacterial infection in humans and mouse [58,59]. These observations were paralleled with the induced expression of *OmDF3b* in spleen of rainbow trout at 24 h after *Y. ruckeri* infection. However, it was found that the expression of both *OmDF1a* and *OmDF1b* were obviously repressed by *Y. ruckeri* infection. In rat (*Rattus norvegicus*), the YTHDF1 knock-down macrophages could protect the brain injury and reduce the inflammation caused by severe sepsis via inhibition of JAK2/STAT3 expression [60]. These observations indicate that the functions of YTHDF1 may differ among species. Obviously, *OmDC1a* showed a relatively high expression abundance in spleen of rainbow trout. The results not only imply a highly active m^6^A methylation in the spleen but also provide evidence for the importance of *OmDC1a* in recognizing the m^6^A modification and mediating its multiple functions. The paralogs of *OmDF1*, *OmDF3* and *OmDC1* in rainbow trout displayed distinct expression patterns in the spleen of rainbow trout after *Y. ruckeri* infection. The results may reflect their more complex regulatory mechanism or functional divergence of m^6^A readers in rainbow trout.

In addition, it was found that the expression levels of *OmDF1b* and *OmDF3b* were significantly changed in both the liver and spleen tissues of rainbow trout under different temperatures or after *Y. ruckeri* infection. They could recognize the m^6^A modification of functional elements or genes to regulate their RNA processing and metabolism in response to environmental temperatures or as a defense against bacterial infection. However, the related biological functions of *OmDF1b* and *OmDF3b* still need to be further studied in the future.

## 4. Materials and Methods

### 4.1. Ethics Statement

All experiments involving animals were conducted according to the proper guidelines and were approved by the respective Animal Research and Ethics Committees of Ocean University of China (Permit number: 20141201). This study did not involve any endangered or protected species, and experiments were performed in accordance with the relevant guidelines.

### 4.2. Identification of YTH Domain-Containing Genes in Rrainbow Trout

To identify the *YTH domain-containing* genes in rainbow trout, the hidden Markov model of the YTH521-B-like domain (PF04146) was downloaded from the Pfam protein database (https://pfam.xfam.org/), where it was used as a query to search against all the amino acid sequences of functional genes in the reference genome (GCF_013265735.2) of rainbow trout using HMMER v3.3.2 software with the default parameters. Then, all available *YTH domain-containing* genes in model organisms, including in human, mouse, and zebrafish, were retrieved from the NCBI database and selected for a reciprocal best hit (RHB) analysis to confirm the accuracy of these candidate *YTH* genes in rainbow trout. Their amino acid properties, molecular weights and theoretical pIs were determined using the online ProtParam tool (https://web.expasy.org/protparam/), and the subcellular localizations were predicted using the Cell-Ploc 2.0 tool (http://www.csbio.sjtu.edu.cn/bioinf/Cell-Ploc-2/ accessed on 18 April 2023) [61].

### 4.3. Phylogenetic Analysis

The phylogenetic analysis was conducted using the amino acid sequences of *YTH domain-containing* genes from rainbow trout, several representative vertebrates and Salmoniformes species, which were retrieved from NCBI database, including human, mouse, chicken, zebrafish, channel catfish, large yellow croaker, Japanese medaka, Nile tilapia, giant grouper, Atlantic salmon, brown trout, chinook salmon, chum salmon, coho salmon, sockeye salmon, and lake trout. Multiple alignments of amino acid sequences were conducted using ClustalW v2.1 software. using the default parameters. The construction of the phylogenetic tree was performed using MEGA v 7.0 software based on the maximum likelihood (ML) method and the Jones–Taylor–Thornton (JTT) model with 1000 bootstrap replicates. The phylogenetic tree was illustrated using Interactive Tree of Life online tool (http://itol.embl.de).

### 4.4. Gene Structure, Chromosome Location and Syntenic Analysis

The exon–intron structures of *YTH domain-containing* genes in rainbow trout were extracted from the general feature format 3 (GFF3) files of the reference genome. The conserved motifs of *YTH domain-containing* genes were predicted using MEME online software (https://meme-suite.org/meme/tools/meme accessed on 18 April 2023). The maximum number of protein motifs was set to 10, and the length of motifs was limited to range from 10 and 50. Then, the results of the gene structure and putative motifs were combined and visualized using Tbtools v1.112 software [62]. In addition, the location information of *YTH domain-containing* genes was also obtained from GFF3 files, which was further compared with their orthologous genes in model organism zebrafish using a syntenic analysis. Reciprocal best hits (RBHs) were performed using BLAST v2.13 software with the threshold of the e-value being <1 × 10^−5^, and it provided strong evidence for the homologs of *YTH domain-containing* genes between zebrafish and rainbow trout. Finally, the visualization of the syntenic analysis was implemented using python-jcvi v1.0.12 software in McScan program. Homologous relationships of duplicated *YTH domain-containing* genes in rainbow trout were visualized using the Circos v0.69.8 program.

### 4.5. Three-Dimensional (3D) Protein Structure Analysis

To investigate the conservation of *YTH domain-containing* genes in evolution, a three-dimensional (3D) structure of proteins encoded by *YTH domain-containing* genes in human genes were first constructed followed their corresponding models in the SWISS-MODEL database (https://swissmodel.expasy.org/). In contrast, the high-accuracy structure of *YTH domain-containing* genes in rainbow trout were predicted and retrieved from the AlphaFold Protein Structure Database, which was created by DeepMind and EMBL’s European Bioinformatics Institute (EMBL-EBI) (https://alphafold.ebi.ac.uk). The amino acid sequences of m^6^A binding pocket sites, which are critical for the specific identification of m^6^A site, were determined according to the published documents [3,63]. The 3D structural models of these *YTH domain-containing* genes were visualized using PyMOL v2.5.4 software.

### 4.6. Temperature Experiment

The temperature experiment was conducted using 480 rainbow trout juveniles (body weight: 91.40 ± 13.63 g) obtained from Wanzefeng Farm (Rizhao, Shandong, China). Prior to the experiment, the fish were randomly selected and transferred to 12 tanks (L × W × H = 1 m × 1 m × 0.6 m, 600 L) at an initial density of 40 individuals/tank for a two-week acclimation. During this period, the water temperature was maintained at 15 ± 0.5 °C using a semi-automatic temperature control system, where the dissolved oxygen level was varied between 6 and 8 mg/L, the pH was varied to range from 7.6 to 7.8, the salinity was kept at 28 ± 0.5, and the photoperiod was 12 h light/12 h dark. After that, the 12 tanks were split into four groups with different temperatures, including 7 °C, 11 °C, 15 °C, and 19 °C. The water temperature was decreased or increased by a steady rate of 1 °C/day towards the targets using a CK-H10 refrigerating machine (Chengke, Guangzhou, China). Thereafter, the temperature was monitored every 2 h using a temperature controller and was held constant for 56 days. The environmental conditions were kept the same as those during the period of acclimation. At the end of the experiment, three individuals in each tank were randomly selected and anesthetized with MS-222. The liver tissue was immediately sampled, quickly frozen in liquid nitrogen and stored at −80 °C until RNA extraction.

### 4.7. RNA Extraction and Quantitative Real-Time PCR (qPCR) Experiment

The total RNA of liver tissues in rainbow trout treated with different temperatures were extracted using the traditional Trizol method. The quality and concentration of the RNA were stimulated and measured using 1.5% agarose gel electrophoresis and a NanoDrop^TM^ One/One^C^ Micro UV-Vis spectrophotometer (Thermo Fisher Scientific, Carlsbad, USA). Then, the high-quality RNA samples were selected for reverse transcription and cDNA synthesis using a HiScript III RT SuperMix for qPCR kit (Vazyme, Nanjing, China). Gene-specific primers were designed using Primer 5 (Premier Biosoft, San Francisco, USA) software, and their information are provided in Table 3. The qPCR experiment was performed using a ChamQ SYBR Color qPCR Master Mix kit (Vazyme, Nanjing, China) and Applied Biosystem QuantStudio^TM^ 5 Real-Time PCR Detection System (Thermo Fisher Scientific, Carlsbad, CA, USA). The 20 μL reaction mixture of qPCR consisted of 10 µL of ChamQ SYBR Color qPCR Master Mix, 0.4 µL of forward primer (10 µM), 0.4 µL of reverse primer (10 µM), 2 µL of template cDNA, and 7.2 µL of ddH_2_O. The amplification procedure for the qPCR reactions contained 3 stages: stage 1, predenaturation (95 °C for 30 s); stage 2, cyclic reaction (40 cycles of 95 °C for 10 s and 60 °C for 30 s); and stage 3, melt curve (95 °C for 15 s, 60 °C for 60 s and 95 °C for 15 s). Three biological replicates and three technical replicates were set up for each experimental treatment. The relative expression levels of each gene were calculated using the 2^−ΔΔCt^ method, and 18s rRNA was used as a housekeeping gene. The values were represented as the mean ± standard error (SE). The data were further analyzed using a one-way analysis of variance (ANOVA) test and Tukey’s multiple comparisons test. The asterisks represent the significant differences between the two groups using Student’s *t*-test.

### 4.8. Bacterial Challenge and Bioinformatic Analysis

To investigate the potential roles of *YTH domain-containing* genes in response to immune challenges, RNA-seq datasets were downloaded from the NCBI Sequence Read Archive (SRA) database under the accession number SRP291881; the datasets were released by Wang et al. in 2021 [64].

The bacterial challenge experiment was conducted as previously described [64]. In brief, *Yersinia ruckeri* bacteria (BH1206 strain) was isolated from infected rainbow trout and cultured in tryptic soy broth medium for 24 h. A total of 36 healthy rainbow trout were collected and obtained from Benxi Agrimarine Industries Inc (Benxi, China). with an average body weight of ~10 g. Prior to the bacterial challenge, these individuals were microscopically and bacteriologically examined to verify the freedom of the *Y. ruckeri* infection. Then, half of the rainbow trout were intraperitoneally injected with 100 µL of *Y. ruckeri* bacteria at 6 × 10^5^ CFU per gram of body weight. The others were regarded as uninfected control fish, which were injected with an equivalent volume of PBS solution. After 24 h post-infection, three individuals in both the control and treatment groups were euthanized with MS-222, quickly dissected for spleen tissue and snap-frozen using liquid nitrogen. The total RNA of the spleen tissue was isolated using the traditional Trizol method, the detailed methods of which have been given in Section 2.7. High-quality libraries were constructed and submitted to the Illumina Hiseq 2500 platform for the paired-end sequencing.

For the bioinformatic analysis, raw reads were filtered and trimmed to remove low-quality reads and adapters using fastp v0.23.2 software. FastQC v0.11.9 software was performed to assess the quality of filtered clean reads, which were then aligned to the reference genome of rainbow trout (USDA_OmykA_1.1) that was downloaded from the NCBI database using HISAT2. software using the default parameters. The counts matrix was constructed based on the alignment file using featureCounts v2.0.1 software. In order to normalize the gene expression levels, the reads counts were converted to FPKM (fragments per kilobase of exon model per million mapped fragments).The DESeq2 R package was used to calculate the average log_2_(FoldChange) and p-value of the expression levels between control and treatment groups.

## 5. Conclusions

In the present study, a complete set of 10 *YTH domain-containing* genes were systematically identified and functionally characterized in rainbow trout. The phylogenetic tree, syntenic relationships, and 3D protein structure analysis provided sufficient evidence for the evolutionary conservation of *YTH domain-containing* genes among distinct species. However, there may be functional divergences among the subfamilies of YTHDF, YTHDC1 and YTHDC2 in rainbow trout, because of the distinct gene sizes, subcellular localizations, exon and intron arrangements, motif organizations, and cage residues. In addition, qPCR experiment indicated that the expression patterns of *OmDF1a*, *OmDF1b*, *OmDF1c*, *OmDF2a*, *OmDF2b*, *OmDF3a*, *OmDF3b*, *OmDC1b*, and *OmDC2* were significantly changed by different temperatures, suggesting their significant contribution to the cellular response to environmental stress. Meanwhile, *OmDF1a*, *OmDF1b*, *OmDF1*, *OmDF3b*, and *OmDC1a* were differentially expressed in spleen tissue of rainbow trout at 24 h after *Y. ruckeri* infection. This study provided a systemic overview of the *YTH domain-containing* genes in rainbow trout and revealed their biological roles in response to abiotic and biotic stresses.

## Figures and Tables

**Figure 1 ijms-24-09348-f001:**
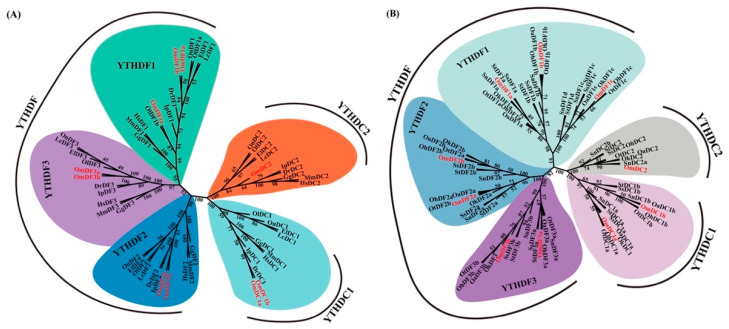
Phylogenetic tree of *YTH domain-containing* genes. (**A**) Phylogenetic tree of representative vertebrates; (**B**) phylogenetic tree of Salmoniformes species. Amino acid sequences of *YTH domain-containing* genes were aligned using ClustalW v2.1 software. The phylogenetic tree was constructed using the ML method and JTT model with 1000 bootstrap replications. *YTH domain-containing* genes in *Oncorhynchus mykiss* are marked with red text. Abbreviations: *YTH domain-containing* genes in *Homo sapiens*, were labeled with Hs; *Mus musculus*, Mm; *Gallus gallus*, Gg; *Danio rerio*, Dr; *Oryzias latipes*, Ol; *Oreochromis niloticus*, On; *Ictalurus punctatus*, Ip; *Larimichthys crocea*, *Lc; Epinephelus lanceolatus*, *El; Salmo salar*, Ss; *Oncorhynchus tshawytscha*, Ot; *Oncorhynchus keta*, Ok; *Oncorhynchus kisutch*, Oh; *Oncorhynchus nerka*, Oa; *Salmo trutta*, St; *Salvelinus namaycush*, Sn.

**Figure 2 ijms-24-09348-f002:**
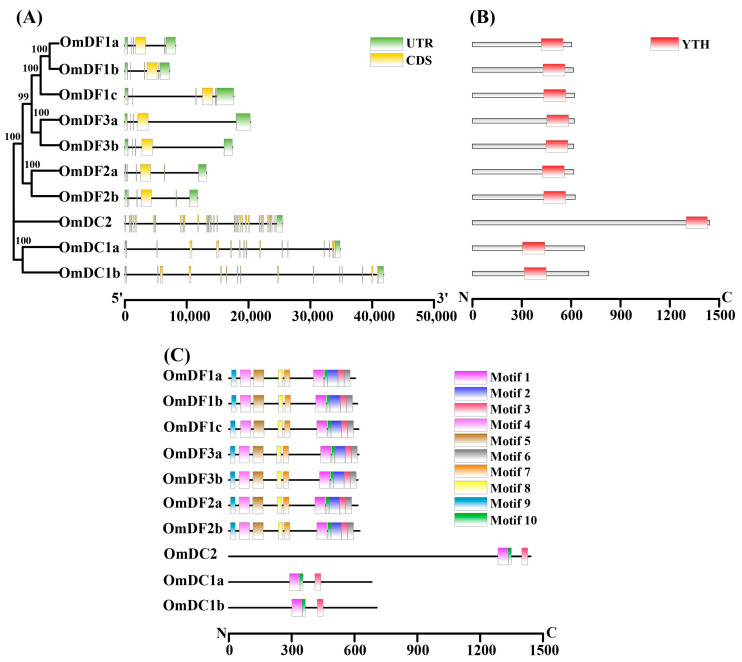
Gene structure and conserved motif analysis of *YTH domain-containing* genes in rainbow trout. (**A**) Exon–intron organizations of *YTH domain-containing* genes. Yellow boxes represent exons, black lines indicate introns, and green boxes show the 5′ or 3′ untranslated regions (UTRs); (**B**) distribution of the YTH domain. The YTH domain is highlighted by red boxes; (**C**) motif composition of YTH proteins. Ten putative motifs were identified and marked with colored boxes.

**Figure 3 ijms-24-09348-f003:**
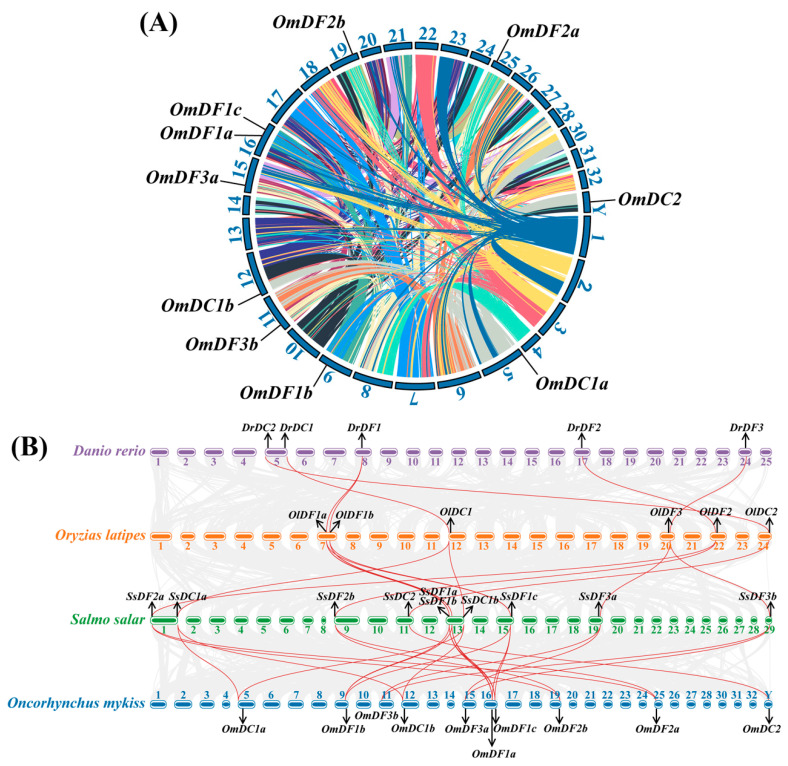
Schematic representations of chromosome locations and synteny relationships of *YTH domain-containing* genes. (**A**) Chromosome locations of *YTH domain-containing* genes in rainbow trout. Paralogous regions were determined using the homologous alignment and are linked by colored lines; (**B**) Genomic synteny of *YTH domain-containing* genes among zebrafish, Japanese medaka, Atlantic salmon, and rainbow trout. The gray lines in the background represent the syntenic blocks of zebrafish and rainbow trout, while homologous *YTH domain-containing* genes are linked by red lines. Abbreviations: *YTH domain-containing* genes in *Danio rerio* were labeled as *Dr*; *Oryzias latipes*, *Ol; Salmo salar*, *Ss*; *Oncorhynchus mykiss*, *Om*.

**Figure 4 ijms-24-09348-f004:**
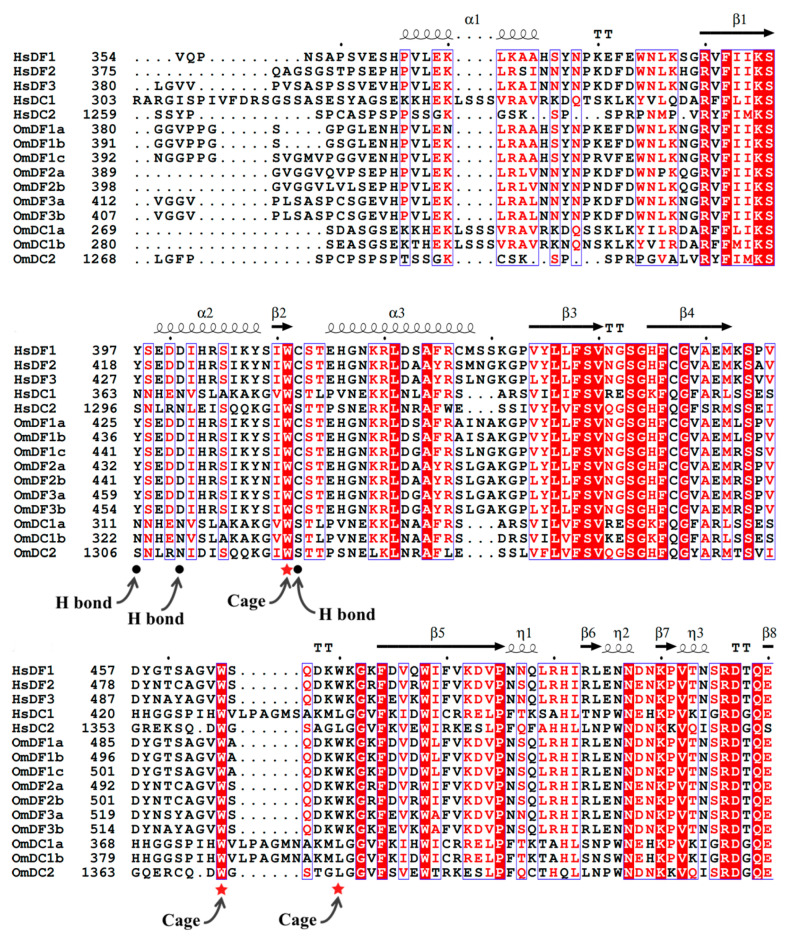
Multiple sequence alignments of YTH domains in humans and rainbow trout. The gene name and amino acid positions are given in the left of the sequence alignments. *YTH domain-containing* genes in humans and rainbow trout were labeled as Hs and Om, respectively. Secondary structural elements were predicted and visualized above the sequence alignment. The medium squiggles represent α-helices, arrows mean β-strands, and TT letters indicate strict β-turns. In the sequence alignments, the red shade with white characters, the strict identical amino acids residues with an identity of 100% were represented by red shade with white characters, while blue frames and mark blocks were used to showing the highly similar sequence [43]. Below the alignments, solid red stars (cage) show the amino acid residues that bind to m^6^A modification. Finally, solid black circles (H bond) represent the residues with hydrogen bonds that interacted with the m^6^A RNA molecule.

**Figure 5 ijms-24-09348-f005:**
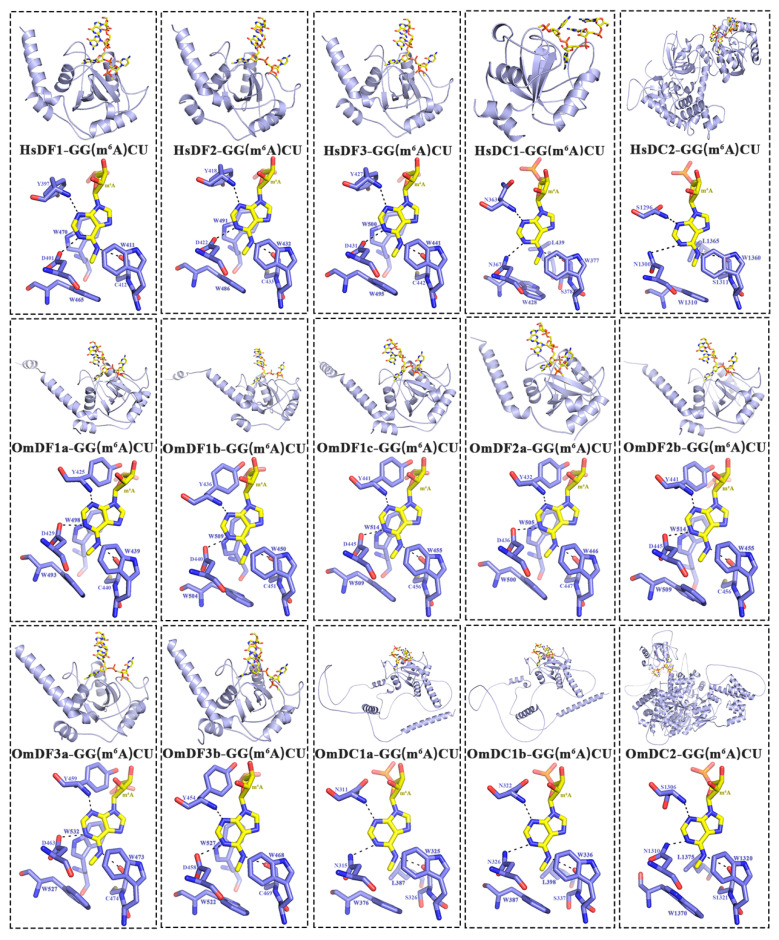
Visualization of three-dimensional (3D) protein structures and YTH-GC (m^6^A) CU complexes in humans and rainbow trout. The secondary structural components (α-helices, β-strands and coils) were shown using cyan colors. In the YTH-GC (m^6^A) CU complexes, the RNA molecules and m^6^A modification were marked in a yellow color, while the amino acid residues of YTH proteins were displayed in a blue color. The hydrogen bonds between the binding cage and m^6^A modification were indicated using black dashed lines.

**Figure 6 ijms-24-09348-f006:**
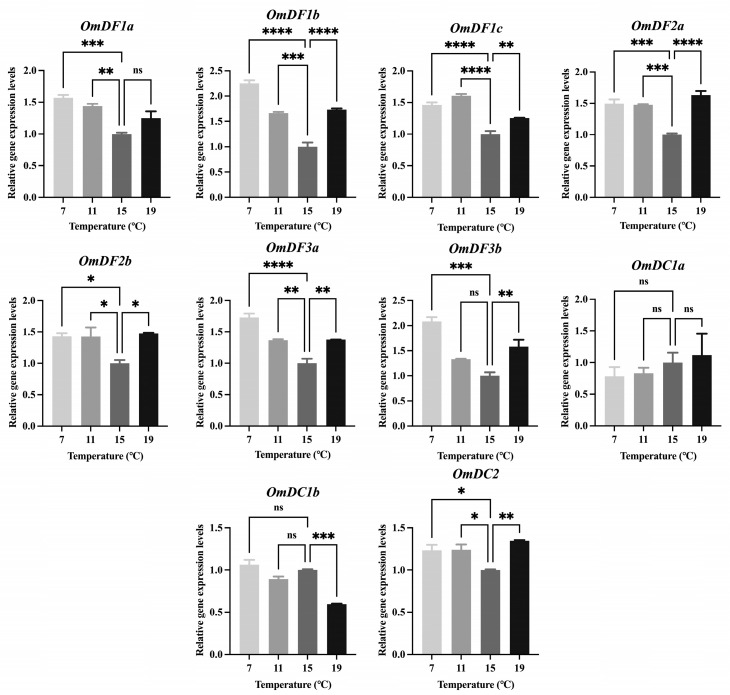
Relative expression levels of *YTH domain-containing* genes in liver tissue of rainbow trout under different temperatures. The expression was calculated using the method of 2−ΔΔCt and normalized using *18S rRNA*. The values were represented as the mean ± S.E. (*n* = 9). The asterisks indicate the significant differences: * *p* < 0.05; ** *p* < 0.01; *** *p* < 0.001; **** *p* < 0.0001. The ‘ns’ represents no significant differences.

**Figure 7 ijms-24-09348-f007:**
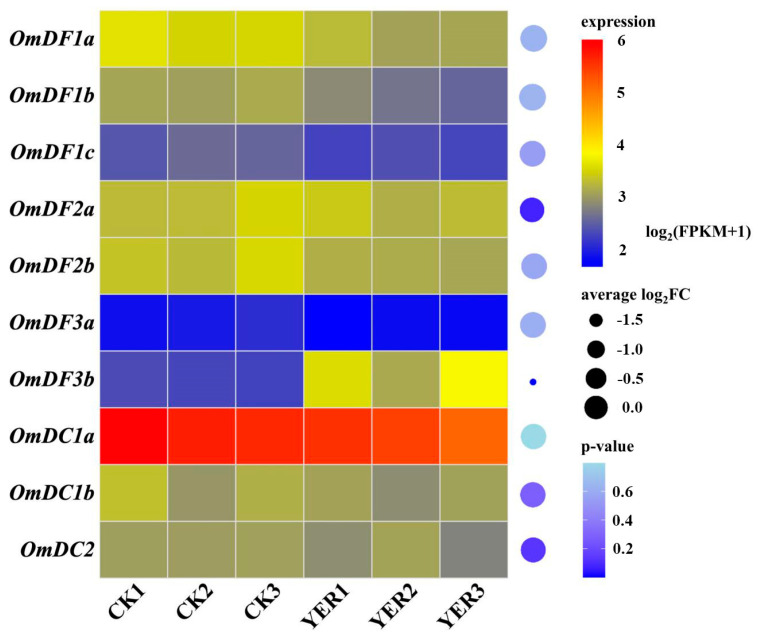
Expression patterns of *YTH domain-containing* genes in the spleens of rainbow trout after 24 h post *Y. ruckeri* infection. CK1-3 represents three replicated RNA-seq datasets of uninfected spleens (control group), while YER1-3 indicates these infected spleens (treatment group). The expression levels were normalized using log_2_ (FPKM+1). The bar on the right indicates the normalized expression data, ranging from low to high (blue to red). The average log_2_(FoldChange) and p-value were determined using DESeq2 R package.

**Table 1 ijms-24-09348-t001:** Characteristics of *YTH domain-containing* gene family in rainbow trout.

Gene Name	Gene ID	Chromosome Position	Gene Length (bp)	ORF (bp)	Exon Number	CDS (aa)	Molecular Weight (kD)	Theoretical pI	Putative Localization		GenBank Accession Number
									Nucleus	Cytoplasm	
*OmDF1a*	LOC110492754	Chr16(−): 46478316–46486506	8191	1812	6	603	65,089.20	7.82	−	+	OQ801191
*OmDF1b*	LOC110532506	Chr09(+): 58975469–58982674	7206	1845	6	614	66,138.59	8.24	−	+	OQ801192
*OmDF1c*	LOC110492496	Chr16(−): 61779118–61796715	17,598	1863	6	620	67,739.79	8.92	−	+	OQ801193
*OmDF2a*	LOC110505278	Chr25(−): 6680033–6693201	13,169	1851	6	616	66,434.05	8.92	−	+	OQ801194
*OmDF2b*	LOC110498162	Chr19(+): 56850905–56862683	11,779	1878	6	625	67,365.05	8.84	−	+	OQ801195
*OmDF3a*	LOC110489655	Chr15(+): 15345480–15365760	20,281	1863	5	620	68,462.39	8.89	−	+	OQ801196
*OmDF3b*	LOC110535333	Chr11(−): 15251271–15268657	17,387	1848	5	615	67,827.79	8.97	−	+	OQ801197
*OmDC1a*	LOC110523074	Chr05(−): 724584–759382	34,799	2049	15	682	78,915.25	7.84	+	−	OQ801198
*OmDC1b*	LOC110536768	Chr12(−): 5222326–5264164	41,839	2124	15	707	81,222.87	9.45	+	−	OQ801199
*OmDC2*	LOC110509841	ChrY(−): 23853359–23878838	25,480	4326	32	1441	160,900.86	8.65	+	−	OQ801200

**Table 2 ijms-24-09348-t002:** List of the putative motifs of *YTH domain-containing* proteins in rainbow trout.

Motif	Length (aa)	Sequence
1	50	YNPKDFDWNLKNGRVFIIKSYSEDDIHRSIKYSIWCSTEHGNKRLD-GAYR
2	50	HFCGVAEMRSPVDYNTSAGVWSQDKWKGKFDVDWLFVKDVPNSQLRHIRL
3	29	ENNDNKPVTNSRDTQEVPLEKAKQVLKII
4	50	MSDPYLPSYYAPSIGFPYSLSEAPWSTGGDPPMPYLPYGQLSNGEH-HFM
5	50	FJYQHGFNFFPENPDFSAWGTSGSQGQSTQSSAYSGSYSYPPSSLG-GAJV
6	29	AGFKHTTSIFDDFSHYEKRQEEEEAVRKT
7	27	GGALPPPPIKHNMDIGTWDNKGSMNKV
8	21	PPKPTSWAAIASKPAKPQPKK
9	23	QRPKGQGNKVQNGSLHQKETVND
10	15	AKGPVYLLFSVNGSG

**Table 3 ijms-24-09348-t003:** Primers of *YTH domain-containing* genes in rainbow trout.

Gene	Primer Sequence (5′ to 3′)
*OmDF1a*	Forward (F): CCAACTCTAGGGACACTCAA Reverse (R): GCTGGCTCATAGGTCTTTCT
*OmDF1b*	Forward (F): TGGACTGGCTGTTTGTGAAAG Reverse (R): TCGGGACCCTCGCTGTATT
*OmDF1c*	Forward (F): ATCCGCCTTGAGAACAACGAC Reverse (R): CCTCCTCTTCCTCCTGCCTCT
*OmDF2a*	Forward (F): CCCCTTTCACCCAGAACGAGC Reverse (R): TGTCCCAAGTGCCGATGTCC
*OmDF2b*	Forward (F): CACCGCTCCATCAAGTACAACATC Reverse (R): TCTCCGCCACGCCACAAA
*OmDF3a*	Forward (F): GTCAACGGCAGTGGTCATTTCT Reverse (R): TTGGGCACGTCCTTCACAAA
*OmDF3b*	Forward (F): GGTGAAGTGGGCGTTTGTGA Reverse (R): AGGTGGTATGCTTGAAAGTGGC
*OmDC1a*	Forward (F): GGCTGGCGGGAAGAAACT Reverse (R): CAAGGTGGGCTGCTTTGG
*OmDC1b*	Forward (F): GGGATGAACGCCAAGATGC Reverse (R): GCGAGGGTCCAGTGTAAAGC
*OmDC2*	Forward (F): CCAAAGCACCCGAACCAC Reverse (R): GAGCACCACGGCACATAACA
*18s rRNA*	Forward (F): ATGGCCGTTCTTAGTTGGTG Reverse (R): TCAGTCTCGTGTGGCTGAAC

## Data Availability

Not applicable.

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
