# Peer review of "Comprehensive Analysis of YTH Domain-Containing Genes, Encoding m6A Reader and Their Response to Temperature Stresses and Yersinia ruckeri Infection in Rainbow Trout (Oncorhynchus mykiss)"

_ijms, 2023, doi:10.3390/ijms24119348_

Round 1
Reviewer 1 Report
The manuscript "Comprehensive analysis of YTH domain genes, coding m6A reader, and their response to thermal stress and infection with Yersinia ruckeri in rainbow trout (Oncorhynchus mykiss)" Yu et al, is interesting and pleasant to read. Fluent and well structured in its parts. In the introduction section, although the authors have ranged in comparing species, they have dedicated little space to the model organism considered. Moreover, in the discussions section, I did not understand the importance of this study in the different research fields where rainbow trout are employed and whether the study results can be translated for humans. Finally, I suggest that tables presented as supplementary material be included in the text where appropriate.
Author Response
Review 1:
Q1: The manuscript "Comprehensive analysis of YTH domain genes, coding m6A reader, and their response to thermal stress and infection with Yersinia ruckeri in rainbow trout (Oncorhynchus mykiss)" Yu et al, is interesting and pleasant to read. Fluent and well structured in its parts. In the introduction section, although the authors have ranged in comparing species, they have dedicated little space to the model organism considered. Moreover, in the discussions section, I did not understand the importance of this study in the different research fields where rainbow trout are employed and whether the study results can be translated for humans. Finally, I suggest that tables presented as supplementary material be included in the text where appropriate.
Response: Thanks for your comments. In the introduction section, we have added several studies of YTH domain-containing genes in model organisms, such as human and mouse (line 52-57). In addition to zebrafish, rainbow trout generally works as another model in scientific research of fishes. The present study has systematically characterized the YTH domain-containing genes in rainbow trout, and investigated their potential roles in response to temperature stress and bacterial infection. The results may serve as a valuable reference for further studies of their biological functions in humans.
In addition, regarding your suggestions, all the supplementary tables have been included in the manuscript.
Introduction (line 72-92):
Because of their importance, YTH domain-containing genes have been systematically defined and extensively studied in higher vertebrates in the last decade. For example, through recognizing the m6A modification of key genes in p38, AKT, ERK1/2, and NF-κB pathways, YTHDF2 could take part in regulating cell injury and apoptosis in lung bronchial epithelial tissue of human [28]. YTHDF1 is able to promote functional axon regeneration in the peripheral nervous system by facilitating the globally translation of injury-induced proteins in dorsal root ganglion of adult mouse [29]. In cattle (Bos taurus), YTHDF1 could recognize the m6A modification in 3’-UTR region of creatine kinase and promote its mRNA translation or stabilization to stimulate the differentiation of bovine skeletal myoblast and regulate the process of myogenesis [30,31]. Additionally, YTH domain-containing genes also play important roles in response to biotic and abiotic stresses. In human, YTHDF2 is proven to participate in regulating the gene expression of HSP90, HSP60, and HSPB1 in response to heat shock stress [32]. It has been reported that the heat shock-inducible lncRNA acts as a transcriptional brake to restain gene expression of heat shock transcription factor 1 and take part in regulating the heat shock response in mouse. However, the heat shock-inducible lncRNA is mediated by the nuclear m6A reader YTHDC1 [33]. In liver of sheep (Ovis aries), The gene expression levels of YTHDC1 and YTHDC2, and the protein abundance of YTHDF3 and YTHDC1 are significantly changed by heat stress, which may be associated with the altered expression of several heat shock proteins (HSP), including HSP70, HSP90 and HSP110 [34]. Similar observation is also reported in bovine animals [35].
References:
- Xiao, K.; Liu, P.; Yan, P.; Liu, Y.; Song, L.; Liu, Y.; Xie, L. N6-methyladenosine reader YTH N6-methyladenosine RNA binding protein 3 or insulin like growth factor 2 mRNA binding protein 2 knockdown protects human bronchial epithelial cells from hypoxia / reoxygenation injury by inactivating p38 MAPK, AKT, ERK1/2, and NF-κB pathways. 2022, 13, 11973-11986. https://doi.org/10.1080/21655979.2021.1999550.
- Yu J.; Li, Y.; Wang, T.; Zhong, X. Modification of N6-methyladenosine RNA methylation on heat shock protein expression. PLoS One. 2018, 13, e0198604. https://doi.org/10.1371/journal.pone.0198604.
- Ji, Q.; Zong, X.; Mao, Y.; Qian, S. B. A heat shock-responsive lncRNA Heat acts as a HSF1-directed transcriptional brake via m6A modification. Natl. Acad. Sci. U. S. A. 2021, 118, e2102175118. https://doi.org/10.1073/pnas.2102175118.
- Lu, Z.; Ma, Y.; Li, Q.; Liu, E.; Jin, M.; Zhang, L.; Wei, C. The role of N6-methyladenosine RNA methylation in the heat stress response of sheep (Ovis aries). Stress. Chaperones. 2019, 24, 333-342. https://doi.org/10.1007/s12192-018-00965-x.
- Qi, Y.; Zhang, Y.; Zhang, J.; Wang, J.; Li, Q. The alteration of N6-methyladenosine (m6A) modification at the transcriptome-wide level in response of heat stress in bovine mammary epithelial cells. Genomics. 2022, 23, 829. https://doi.org/10.1186/s12864-022-09067-6.
Q2: In the present manuscript entitled 'Comprehensive analysis of YTH-domain genes, encoding m6A 2 reader, and their response to temperature stresses and Yersinia ruckeri infection in rainbow trout (Oncorhynchus mykiss)', the authors present a detailed analysis of YTH/domain genes in O. mykiss. This can be helpful in studying m6A modifications in RNA. However, a few corrections are necessary in the current form of this manuscript.
Response: Thanks for valuable comments. We have carefully checked the manuscript twice and made some corrections as requested.

Reviewer 2 Report
In the present manuscript entitled 'Comprehensive analysis of YTH-domain genes, encoding m6A 2 reader, and their response to temperature stresses and Yersinia ruckeri infection in rainbow trout (Oncorhynchus mykiss)', the authors present a detailed analysis of YTH/domain genes in O. mykiss. This can be helpful in studying m6A modifications in RNA. However, a few corrections are necessary in the current form of this manuscript.
Introduction:
Line 51: Overall, the conservation of m6A-binding properties of YTH domain-containing proteins in different eukaryotic species suggests that m6A modification and its regulation by these proteins have important functions that have been maintained throughout evolution. It would be nice to mention a few examples related to the evolutionary conservation of YTH domain-containing genes. Please cite relevant papers.
Line 76: Although a lot has been discussed here regarding recent studies of YTH-domain genes, this section lacks information related to the link between the role of temperature stress in other organisms (including mammals) and vertebrates. It would be nice to include a few lines on the molecular pathway of temperature stress and YTH-domain genes and whether they are similar in vertebrates.
Results:
Line 123: The phylogenetic analysis in this section is quite informative related to the evolutionary concept of YTH genes. However, this shows only a comparison in a diverse group of species. It would be nice to include another phylogenetic analysis within Salmoniformes species (a lot of predicted sequences are available on NCBI).
Line 203: It will be important to show how these genes are conserved within the Salmoniformes order.
Line 211: Is there any specific reason to compare human and O. mykiss multiple sequence alignment? Why not with any closely related vertebrates?
Line 244: As 15 degrees Celsius is optimum for O. mykiss growth, then it should be taken as a control for qPCR experiments.
Line 258: A few statistics obtained from RNA-seq data analysis should be mentioned.
Discussion:
Line 318: Why does OmDC1a have no significant expression in temperature stress? What is its significance in other organisms related to temperature stress?
Line 334: It will be important to discuss in a few lines the molecular pathways through which these YTH domain genes affect temperature stress and bacterial infection. Is there any common link between them?
Methods:
Line 380: Neighbor joining is just a clustering algorithm that clusters haplotypes based on genetic distance and is not often used for publication in recent literature. Moreover, long branch attraction (LBA) artifacts might occur when distantly related taxa evolve at a similar rate, causing them to cluster together in the phylogenetic tree, resulting in an incorrect grouping. Why did the authors choose this method over maximum likelihood? Check the results with the ML method tree.
Line 418: According to some studies, rainbow trout raised in controlled environments at a temperature of 25 degrees Celsius can achieve an average weight gain of about 1.5 to 2.0 grams per day. Why 7℃, 11℃, 15℃ and 19℃? Is there any specific reason to choose these temperatures?
Line 448: As this data is retrieved from a previous study, this should be explained properly. The 2nd paragraph sounds like this has been done in the present study. This should be rewritten citing original authors properly.
